# Modulation of FLT3-ITD Localization and Targeting of Distinct Downstream Signaling Pathways as Potential Strategies to Overcome FLT3-Inhibitor Resistance

**DOI:** 10.3390/cells10112992

**Published:** 2021-11-03

**Authors:** Maximilian Fleischmann, Mike Fischer, Ulf Schnetzke, Colin Fortner, Joanna Kirkpatrick, Florian H. Heidel, Andreas Hochhaus, Sebastian Scholl

**Affiliations:** 1Klinik für Innere Medizin II, Abteilung Hämatologie und Onkologie, Universitätsklinikum Jena, 07747 Jena, Germany; Mike.Fischer@med.uni-jena.de (M.F.); Ulf.Schnetzke@med.uni-jena.de (U.S.); colin.fortner@uni-jena.de (C.F.); florian.heidel@uni-greifswald.de (F.H.H.); Andreas.Hochhaus@med.uni-jena.de (A.H.); 2Department for Proteomic Analysis, The Francis Crick Institute, London NW1 1AT, UK; jokirkpatrick@hotmail.com; 3Leibniz Institute on Aging–Fritz-Lipmann-Institute, 07745 Jena, Germany; 4Klinik für Innere Medizin C, Universitätsmedizin Greifswald, 17489 Greifswald, Germany

**Keywords:** AML, FLT3-ITD, HSP90, tunicamycin, 17-AAG, VPA, rapamycin, resistance, TKI

## Abstract

OBJECTIVES: Internal tandem duplications (ITDs) of the Fms-like tyrosine kinase 3 (FLT3) represent the most frequent molecular aberrations in acute myeloid leukemia (AML) and are associated with an inferior prognosis. The pattern of downstream activation by this constitutively activated receptor tyrosine kinase is influenced by the localization of FLT3-ITD depending on its glycosylation status. Different pharmacological approaches can affect FLT3-ITD-driven oncogenic pathways by the modulation of FLT3-ITD localization. AIMS: The objective of this study was to investigate the effects of N-glycosylation inhibitors (tunicamycin or 2-deoxy-D-glucose) or the histone deacetylase inhibitor valproic acid (VPA) on FLT3-ITD localization and downstream activity. We sought to determine the potential differences between the distinct FLT3-ITD variants, particularly concerning their susceptibility towards combined treatment by addressing either N-glycosylation and the heat shock protein 90 (HSP90) by 17-AAG, or by targeting the PI3K/AKT/mTOR pathway by rapamycin after treatment with VPA. METHODS: Murine Ba/F3 leukemia cell lines were stably transfected with distinct FLT3-ITD variants resulting in IL3-independent growth. These Ba/F3 FLT3-ITD cell lines or FLT3-ITD-expressing human MOLM13 cells were exposed to tunicamycin, 2-deoxy-D-glucose or VPA, and 17-AAG or rapamycin, and characterized in terms of downstream signaling by immunoblotting. FLT3 surface expression, apoptosis, and metabolic activity were analyzed by flow cytometry or an MTS assay. Proteome analysis by liquid chromatography–tandem mass spectrometry was performed to assess differential protein expression. RESULTS: The susceptibility of FLT3-ITD-expressing cells to 17-AAG after pre-treatment with tunicamycin or 2-deoxy-D-glucose was demonstrated. Importantly, in Ba/F3 cells that were stably expressing distinct FLT3-ITD variants that were located either in the juxtamembrane domain (JMD) or in the tyrosine kinase 1 domain (TKD1), response to the sequential treatments with tunicamycin and 17-AAG varied between individual FLT3-ITD motifs without dependence on the localization of the ITD. In all of the FLT3-ITD cell lines that were investigated, incubation with tunicamycin was accompanied by intracellular retention of FLT3-ITD due to the inhibition of glycosylation. In contrast, treatment of Ba/F3-FLT3-ITD cells with VPA was associated with a significant increase of FLT3-ITD surface expression depending on FLT3 protein synthesis. The allocation of FLT3 to different cellular compartments that was induced by tunicamycin, 2-deoxy-D-glucose, or VPA resulted in the activation of distinct downstream signaling pathways. Whole proteome analyses of Ba/F3 FLT3-ITD cells revealed up-regulation of the relevant chaperone proteins (e.g., calreticulin, calnexin, HSP90beta1) that are directly involved in the stabilization of FLT3-ITD or in its retention in the ER compartment. CONCLUSION: The allocation of FLT3-ITD to different cellular compartments and targeting distinct downstream signaling pathways by combined treatment with N-glycosylation and HSP90 inhibitors or VPA and rapamycin might represent new therapeutic strategies to overcome resistance towards tyrosine kinase inhibitors in FLT3-ITD-positive AML. The treatment approaches addressing N-glycosylation of FLT3-ITD appear to depend on patient-specific FLT3-ITD sequences, potentially affecting the efficacy of such pharmacological strategies.

## 1. Introduction

Acute myeloid leukemia (AML) patients with high-risk genetics are associated with a poor outcome (ELN) [1].

Mutations of FLT3–a class I receptor tyrosine kinase represent the most frequent molecular aberrations that are detected in patients with newly diagnosed AML [2]. FLT3 mutations are classified into FLT3 length mutations (internal tandem mutations (ITDs)) and FLT3 mutations that are located in the second tyrosine kinase domain (FLT3-TKD mutations) [3,4,5]. FLT3-ITDs represent the majority of FLT3 mutations and are either located in the juxtamembrane (JM) region (about 70%) or in the TKD1 domain (about 30%) [6,7]. 

There is a broad variety of FLT3-ITD mutations that result in patient-specific localization, length, and peptide sequences of the mutated gene [8]. While all FLT3-ITDs mediate a ligand-independent and constitutive activation of FLT3, the localization of ITDs (JM region versus TKD1 domain) is supposed to define the differences in FLT3 downstream signaling and may affect the level of sensitivity towards distinct tyrosine kinase inhibitors [9,10,11].

FLT3 tyrosine kinase inhibitors (FLT3i) comprise a broad spectrum of distinct compounds that have been evaluated in clinical studies [12]. In detail, the first generation inhibitor midostaurin has been investigated in combination with induction chemotherapy of AML patients in the RATIFY trial leading to its approval for newly diagnosed AML patients with activating FLT3 mutation [13]. For patients with relapsed or refractory AML (r/r AML) harboring FLT3-ITD mutations, the second generation FLT3i quizartinib was able to improve survival compared with the standard chemotherapy regimens [14]. Recently, the third generation FLT3i gilteritinib has been approved for patients with r/r AML, demonstrating a clinically relevant survival advantage in those AML patients that harbor either FLT3-ITD or FLT3-TKD mutations [15].

While primary resistance towards FLT3i is rarely observed in AML patients with activating FLT3 mutations, treatment regularly induces secondary resistance, representing a limitation of this therapeutic approach. A key molecular mechanism of FLT3i resistance is represented by so-called “on target” resistance that is mediated by additional mutations of FLT3 [16,17]. In contrast, recent reports describe potential “off target” mechanisms independently mediating FLT3i resistance (e.g., by constitutive activation of other oncogenic signaling pathways) [18,19].

Depending on the underlying molecular mechanism, distinct strategies are necessary to overcome clinical resistance towards FLT3i. A potential approach to address “on target” resistance after FLT3i treatment is the modulation of cellular localization of FLT3-ITD in AML cells. The FLT3 wild-type protein is mainly localized within the cellular membrane while trafficking is directed by stepwise glycosylation within the endoplasmic reticulum (ER) and Golgi apparatus (GA). In detail, FLT3 undergoes a complex glycosylation process resulting in the complex glycosylated 150 kDa FLT3 protein. In contrast, FLT3-ITD is significantly retained in the ER/GA system which is attributed to an impaired post-translational processing and reduced glycosylation of the immature 130 kDa FLT3-ITD protein [20]. Intracellular localization of FLT3-ITD defines which downstream signaling pathways are constitutively activated by FLT3-ITD. In detail, FLT3-ITD that is located in the cell membrane predominantly activates AKT and MEK while FLT3-ITD that is retained in the ER/GA system predominantly activates the STAT5 signaling pathway [21].

There are several experimental approaches to inhibit N-glycosylation of FLT3-ITD: first, tunicamycin represents a mixture of nucleoside antibiotics that are able to block the transfer of sugar residues to dolicholphosphate being indispensable for N-glycosylation; second, 2-deoxy-D-glucose that preferentially depletes ATP can also inhibit N-glycosylation; and third, the depletion of dolicholphosphate by the clinically applied drug Fluvastatin [22,23,24]. The treatment of FLT3-ITD-positve AML cells with tunicamycin at low concentrations in combination with FLT3i shows synergistic effects with respect to the induction of apoptosis [25].

Heat shock protein 90 (HSP90) is one of the key chaperones that is responsible for protein stabilization of oncogenic tyrosine kinases and FLT3-ITD represents an important client kinase of HSP90. FLT3-ITD is released from HSP90 upon treatment with a HSP90 inhibitor (e.g., 17-AAG) thus leading to degradation of FLT3-ITD. In contrast to the wild-type FLT3 protein, the HSP90 client kinase FLT3-ITD shows a stabilization from proteomic degradation HSP [26]. The inhibition of such important chaperone proteins as HSP90 can induce increased apoptosis in AML cells with FLT3-ITD mutations and might contribute to overcome FLT3i resistance [27,28].

While the impact of such histone deacetylase inhibitors (HDACi), as valproic acid (VPA), on differentiation induction of AML cells has been intensively analyzed, less is known about the potential non-genomic effects of HDACi [29,30]. VPA has been investigated in a single experimental study of FLT3-ITD positive cells. In detail, the combination of all-trans retinoic acid (ATRA) and VPA was able to increase the activation of the phosphatidyl-3 kinase (PI3K)/AKT signaling pathway and was associated with a higher susceptibility towards the mammalian target of rapamycin (mTOR) inhibition [31].

Here, we sought to investigate different molecular mechanisms that might contribute to overcoming FLT3i resistance. We addressed the question of whether VPA can affect both FLT3-ITD localization and constitutively activated downstream signaling, respectively. In addition, we analyzed the impact of HSP90 inhibition on FLT3-ITD after its entrapment in the ER/Golgi system by tunicamycin or 2-deoxy-D-glucose.

## 2. Materials and Methods

### 2.1. Cell Lines and Growth Conditions

The human cell line MOLM13 harboring monoallelic FLT3-ITD was purchased from the “Deutsche Sammlung von Mikroorganismen und Zellkulturen GmbH” (DSMZ, Braunschweig, Germany). In addition, four murine Ba/F3 cell lines that were harboring different FLT3-ITD variants either in the juxtamembrane domain (JMD) or in the tyrosine kinase domain 1 (TKD1) were used. The location and length of different FLT3-ITD variants are illustrated in Appendix A. The Ba/F3 KRAS (G12D) cell line served as a negative control for potential “off target” effects in the Ba/F3-ITD cell lines. All Ba/F3 cell lines had been stably transfected with either distinct FLT3-ITD constructs or with oncogenic KRAS (G12D) as previously described [10], resulting in an Il-3 independent growth. The cells were maintained in RPMI 1640 medium that was supplemented with 10% fetal calf serum + 2 mmol/L L-glutamine (Gibco BRL, Wiesbaden, Germany) in a 5% CO_2_ air environment and 37 °C fully humidified incubator. The day before starting an experiment, the cells were supplied with fresh culture medium.

### 2.2. Reagents

The HDAC inhibitor valproic acid (VPA) and 2-deoxyglucose were purchased from Sigma-Aldrich (Munich, Germany) and dissolved in water. Tunicamycin, all-trans retinoic acid (ATRA), tanespimycin (17-AAG), and cycloheximide (CHX) were purchased from Sigma-Aldrich as well, rapamycin from Selleck Chemicals and were all dissolved in dimethyl-sulphoxide (DMSO). All of the reagents were stored at −20 °C in aliquots prior to use.

### 2.3. Antibodies

The following antibodies were used for immunoblotting: anti-phospho-STAT5 (#9351), anti-phospho-AKT (#9271), anti-phospho-ERK1/2 (#9370), anti-AKT (#4685), anti-ERK1/2 (#9102), anti-MCL1 (#5453), anti-histone H3 (#9717), and anti-acetylated histone H3 (#9677). All antibodies were all acquired from Cell Signaling Technology, Frankfurt, Germany. Anti-FLT3 (sc-480), anti-acetyl histone h4 (sc-377520) anti-beta actin (sc-8432), anti-Hsp90 (sc-101494), anti-STAT5 (sc-835), anti-GAPDH (sc-20357), and anti-goat IgG HRP secondary antibody (sc-2345) were purchased from Santa Cruz Biotechnology, Heidelberg, Germany while anti-rabbit and anti-mouse IgG HRP secondary antibody were purchased from Promega, Madison, WI, USA.

For flow cytometry, the anti-CD135 (#313305), anti-Annexin V (#640920), and human IgG1 PE isotype control (#403504) were purchased from BioLegend, San Diego, CA, USA.

### 2.4. RNA Preparation and qRT-PCR Analysis

Total RNA was isolated using the innuPREP RNA Mini Kit (Analytik Jena, Jena, Germany) according to standard protocol. First-strand cDNA synthesis was performed with 1 μg of total RNA using M-MLV reverse transcriptase according to manufacturer’s protocol (Invitrogen, Karlsruhe, Germany). SYBR green-based analyses were performed using the Mastercycler^®^ ep realplex Real-time PCR System (Eppendorf, Hamburg, Germany). The reaction set up (20 μL) was as follows: 20 ng cDNA, 0.5 μM of each primer, and 1× FastStart SYBR Green Master (Roche, Mannheim, Germany). The following primer sets were used: FLT3: 5′-TCTCAATCCAGGTTGCCGTC-3′ and 5′-AAATTGGTCCTGACAGTGTGC-3′; Actin: 5′-AGCCATGTACGTAGCCATCC-3′ and 5′-CTCTCAGCTGTGGTGGTGAA-3′. All of the reactions were carried out at 95 °C for 10 min following 40 cycles at 95 °C for 10 s, 58 °C for 15 s, and 72 °C for 20 s.

### 2.5. MTS Assay and PrestoBlue Assay

The metabolic activity of Ba/F3 and MOLM13 cells was analyzed using 3-(4,5-dimethylthiazol-2-yl)-2,5-diphenyltetrazolium bromide (MTS, Promega, Madison, WI, USA). In detail, 20 µL MTS reagent was added to the cell suspension in 96-well plates. After centrifugation and incubation for 1 to 4 h, readouts were performed with the Clariostar plate-reader and analyzed by Mars Data Analysis Software (BMG Labtech, Ortenberg, Germany). As an additional tool for comparing cell viability, PrestoBlue reagent (Thermo Fisher Scientific, Waltham, MA, USA) was used. After incubation of the 90 µl of cell suspension in 96-well plates, 10 µL of PrestoBlue reagent were added and analysis was performed after 30 min incubation by the Clariostar plate-reader. The data were processed Mars Data Analysis Software (BMG Labtech).

### 2.6. Protein Isolation for Western Blot and Proteome Analysis

After incubation of Ba/F3 or MOLM13 cells under the different conditions, the cells were harvested and washed three times and resuspended in serum-free medium. The preparation of cell lysates was performed according to standard protocols using a radioimmunoprecipitation assay buffer (RIPA) that was supplemented with fresh protease inhibitors and sodium orthovanadate. The protein lysates were subjected to sodium dodecyl sulphate-polyacrylamide gel electrophoresis (SDS-PAGE) and blotted onto a polyvinylidene fluoride (PVDF) membrane before blocking for 1 h at room temperature. Next, the primary antibody was incubated overnight at 4 °C or for 1 h in a 37 °C warmed water-bath. After incubation with the secondary antibody for 1 h at room temperature, the membrane was visualized by adding enhanced chemiluminescence reagent. Digital images were taken using the Fujifilm LAS-3000 (Düsseldorf, Germany) and Vilber Fusion FX and densitometry was quantified by ImageJ Software. Sample preparation, sample processing, and data analysis of whole proteome analysis by means of liquid chromatography-mass spectrometry (LC-MS/MS) is comprehensively described in the Appendix A.

### 2.7. Apoptosis Assay

For the Annexin V apoptosis assay, 5 × 10^5^ cells/mL were seeded in 6-, 12-, or 24-well plates and different concentrations of the indicated compounds were added. DMSO 0.1% (*v*/*v*) served as a negative control. The plates were incubated for different times at 37 °C and before the samples were washed twice with cold phosphate-buffered saline (PBS). The cell pellet of each sample was resuspended in 50 μL of staining solution containing 0.125 μL Annexin V-allophycocyanin (APC) and 49 μL Annexin V binding buffer and was incubated for 30 min in the dark at room temperature. After the addition of 400 μL Annexin V binding buffer per sample, the cells were analyzed by flow cytometry using a FACSCalibur (Becton Dickinson, Heidelberg, Germany).

### 2.8. CD135 Analyses by Flow Cytometry

FLT3 surface expression (CD135) was detected by extracellular staining and subsequent flow cytometric analysis. In detail, 5 × 10^5^ cells/mL were seeded in 6-, 12-, or 24-well plates and incubated for the indicated conditions at 37 °C. Next, all samples were washed twice with medium before the anti-CD135 primary antibody was added in relation 1:20 and incubated for 30 min in the dark at room temperature. After repeated washes, the samples were resuspended in 400 μL buffer solution for measurement on a FACSCalibur and analyzed with CellQuest^TM^ software (Becton Dickinson). The signals were averaged using the geometric mean and defined as mean fluorescence intensity (MFI). An isotype control IgG staining for each sample was performed to calculate background MFI.

### 2.9. Data Analysis and Statistical Analysis

A two-tailed Student’s *t*-test was used for statistical analyses, *p* values < 0.05 were considered statistically significant (**** *p* < 0.0001, *** *p* < 0.001, ** *p* < 0.01, * *p <* 0.05, *ns* not significant). The statistical analyses were performed using Prism 6 Software (GraphPad Software, Inc., La Jolla, CA, USA). Unless otherwise indicated, the data points represent the mean value and standard deviation of three biological triplicates. The data of flow cytometry experiments were analyzed by FlowJo Software Version 9. IC_50_ values based on viability data that were measured by the MTS assay were calculated with non-linear regression models in GraphPad. Additionally, the raw data were processed to SynergyFinder Software to investigate additional or synergistical effects for combination of compounds [32]. Additional effects were assumed when the synergy score (Bliss or ZIP) was between 0 and 10 and the synergistic effects for 10 and higher.

### 2.10. Proteome Analysis

#### 2.10.1. Sample Preparation-Lysis, Acetone Precipitation, Digestion and Clean-Up

The cell pellets were resuspended in lysis buffer (final concentration: 0.1 M HEPES/pH 8; 2% SDS; 0.1 M DTT), vortexed, and then sonicated using a Bioruptor (Diagenode) (10 cycles with 1 min on and 30 s off with high intensity @ 20 °C). For reduction full denaturation of the proteins, the lysates were first incubated at 95 °C for 10 min and subsequently sonicated in the Bioruptor for a further 10 cycles as before. The supernatants (achieved after centrifugation, 20,000× *g*, room temperature, 2 min) were then treated with iodacetamide (room temperature, in the dark, 30 min, 200 mM). After running a coomassie gel to estimate the amount of protein in each sample, approximately 50 µg of each sample was treated with 4 volumes ice cold acetone to 1 volume sample and left overnight at −20 °C to precipitate the proteins. The samples were then centrifuged at 20,000× *g* for 30 min, 4 °C. After removal of the supernatant, the precipitates were washed twice with 400 µL 80% acetone (ice cold). After each wash step, the samples were vortexed, then centrifuged again for 2 min at 4 °C. The pellets were then allowed to air-dry before being dissolved in a digestion buffer (60 µL, 3 M urea in 0.1 M HEPES, pH 8) with sonication (3 cycles in the Bioruptor as above) and incubated for 4 h with LysC (1:100 enzyme: protein ratio) at 37 °C with shaking at 600 rpm. The samples were diluted 1:1 with milliQ water (to reach 1.5 M urea) and were incubated with trypsin (1:100 enzyme:protein ratio) for 16 h at 37 °C. The digests were then acidified with 10% trifluoroacetic acid and then desalted with Waters Oasis^®^ HLB µElution Plate 30 µm in the presence of a slow vacuum. In this process, the columns were conditioned with 3 × 100 µL solvent B (80% acetonitrile; 0.05% formic acid) and equilibrated with 3 × 100 µL solvent A (0.05% formic acid in milliQ water). The samples were loaded, washed 3 times with 100 µL solvent A, and then eluted into PCR tubes with 50 µL solvent B. The eluates were dried down with the speed vacuum centrifuge and dissolved in 50 µL 5% acetonitrile, 95% milliQ water, with 0.1% formic acid prior to analysis by LC-MS/MS.

#### 2.10.2. LC-MS/MS

The peptides were separated using the nanoAcquity UPLC system (Waters) fitted with a trapping (nanoAcquity Symmetry C_18_, 5 µm, 180 µm × 20 mm) and an analytical column (nanoAcquity BEH C_18_, 1.7 µm, 75 µm × 250 mm). The outlet of the analytical column was coupled directly to an Orbitrap Fusion Lumos (Thermo Fisher Scientific) using the Proxeon nanospray source. Solvent A was water and 0.1% formic acid and solvent B was acetonitrile and 0.1% formic acid. The samples (500 ng) were loaded with a constant flow of solvent A at 5 µL/min onto the trapping column; the trapping time was 6 min. The peptides were eluted via the analytical column with a constant flow of 0.3 µL/min. During the elution step, the percentage of solvent B increased in a linear fashion from 3% to 25% in 30 min, then increased to 32% in 5 more minutes, and finally to 50% in a further 0.1 min. The total runtime was 60 min. The peptides were introduced into the mass spectrometer via a Pico-Tip Emitter 360 µm OD × 20 µm ID, 10 µm tip (New Objective), and a spray voltage of 2.2 kV was applied. The capillary temperature was set at 300 °C. The RF lens was set to 30%. Full scan MS spectra with mass range of 375–1500 *m/z* was acquired in profile mode in the Orbitrap with resolution of 120,000. The filling time was set at maximum of 50 ms with limitation of 2 × 10^5^ ions. The “Top Speed” method was employed to take the maximum number of precursor ions (with an intensity threshold of 5 × 10^3^) from the full scan MS for fragmentation (using HCD collision energy, 30%), quadrupole isolation (1.4 Da window), and measurement in the ion trap, with a cycle time of 3 s. The MIPS (monoisotopic precursor selection) peptide algorithm was employed but with relaxed restrictions when too few precursors that met the criteria were found. The fragmentation was performed after the accumulation of 2 × 10^3^ ions or after a filling time of 300 ms for each precursor ion (whichever occurred first). MS/MS data was acquired in centroid mode, with the rapid scan rate and a fixed first mass of 120 *m/z*. Only multiply charged (2^+^–7^+^) precursor ions were selected for MS/MS. Dynamic exclusion was employed with a maximum retention period of 60 s and a relative mass window of 10 ppm. Isotopes were excluded. Additionally, only 1 data dependent scan was performed per precursor (only the most intense charge state was selected). The ions were injected for all available parallelizable time. To improve the mass accuracy, a lock mass correction using a background ion (*m/z* 445.12003) was applied. Data acquisition was performed using Xcalibur 4.0/Tune 2.1 (Thermo Fisher Scientific).

#### 2.10.3. Data Analysis

For the quantitative label-free analysis, the raw files from the Orbitrap Fusion Lumos were analyzed using MaxQuant (version 1.5.3.28) [33]. The MS/MS spectra were searched against the *Mus Musculus* Swiss-Prot entries of the Uniprot KB (database release 2016_01, 16,755 entries) using the Andromeda search engine [34]. A list of common contaminants was appended to the database search. The search criteria were set as follows: full tryptic specificity was required (cleavage after lysine or arginine residues, unless followed by proline); 2 missed cleavages were allowed; oxidation (M), and acetylation (protein N-term) were applied as variable modifications, with mass tolerances of 20 ppm set for precursor and 0.5 Da for fragments. The reversed sequences of the target database were used as a decoy database. The peptide and protein hits were filtered at a false discovery rate of 1% using a target-decoy strategy [35]. Additionally, only proteins that were identified by at least 2 unique peptides were retained. The LFQ intensity values per protein (from the proteinGroups.txt output of MaxQuant) were used for further analysis. All comparative analyses were performed using R version 3.2.3 [36]. The data was quantile normalized to reduce technical variations. The protein differential expression between the pairwise conditions evaluated was carried out using the Limma package [37]. The differences in protein abundances were statistically determined using the Student’s *t*-test with variances that were moderated by *limma*’s empirical Bayes method. A false discovery rate was estimated using fdrtoo.l [38].

## 3. Results

### 3.1. Signal Transduction, Apoptosis, and Metabolic Activity after Treatment of Ba/F3 Cell Lines Expressing Four Distinct FLT3-ITD Variants with 17-AAG after Pre-Incubation with Tunicamycin

We analyzed the impact of four distinct FLT3-ITD variants on the susceptibility to the HSP90 inhibitor, 17-AAG. A total of four different Ba/F3 cell lines, stably expressing different FLT3-ITD isoforms (598/599, G613E, FV605YV or E611V) were either pre-treated with tunicamycin or DMSO prior to incubation with 17-AAG at several concentrations ranging from 125 to 1000 nM (Figure 1).

The inhibition of protein glycosylation with tunicamycin prior to 17-AAG treatment resulted in a concentration-dependent reduction of phosphorylated STAT5 that can be observed in all four FLT3-ITD cell lines. In detail, there is a slight decrease of phosphorylated STAT5 with increasing concentrations of 17-AAG that is most pronounced in the Ba/F3 cells that were expressing the FLT3-ITD variant G613E.

Single treatment with 17-AAG did not affect the expression of total STAT5 in all four analyzed Ba/F3 cell models. Furthermore, pre-treatment with tunicamycin was associated with an only slight decrease of STAT5 expression. 17-AAG was mediating significant decreases of phosphorylated STAT5 after pre-treatment with tunicamycin that were observed within a concentration range of 17-AAG that was not affecting the protein level of total STAT5.

In contrast, the expression of FLT3 and total AKT was significantly more reduced after sequential treatment with tunicamycin and 17-AAG as compared to single treatment with 17-AAG. The concentration-dependent phosphorylation of ERK following treatment with 17-AAG alone or in combination with tunicamycin did not alter the protein expression of total ERK. A similar observation as with the phosphorylated STAT5 was seen with the phosphorylated ERK that is significantly more inhibited by 17-AAG when Ba/F3 FLT3-ITD cells are pre-incubated with tunicamycin. In summary, dose-dependent reductions of phosphorylated signaling pathways was much more pronounced when the cells were preincubated with tunicamycin.

The analysis of cellular metabolic activity using the MTS assay and measurement of apoptosis by means of the Annexin V assay, respectively, demonstrated a significant impact of pre-incubation with tunicamycin prior to 17-AAG treatment in all analyzed Ba/F3 FLT3-ITD cell lines. Importantly, the comparison of responses at lower dose levels of 17-AAG–i.e., 125 nM, and especially 250 nM, respectively, significantly differed between the four distinct FLT3-ITD variants that were stably expressed by Ba/F3 cells. Pre-treatment of Ba/F3-G613E cells harboring the ITD mutation within the TKD1 domain with tunicamycin followed by incubation of 17-AAG at a concentration of 250 nM resulted in a decrease of Annexin V-negative cells to 32.8% as compared to 82.8% (*p* = 0.0006) when Ba/F3-G613E cells were pre-incubated with 0.1% (*v*/*v*) DMSO. A similar observation was made in the Ba/F3-FV613YF cells (ITD mutation within the JM domain): 52.5% versus 85.6% (*p* = 0.0025), respectively. In contrast, the response to sequential treatment with tunicamycin and 17-AAG was only moderate and more pronounced at higher concentrations of 17-AAG in Ba/F3-598/599 cells (JM domain) and Ba/F3-E611V cells (TKD1 domain), respectively. The complete analyses of MTS and Annexin V assay is summarized in the Appendix A. By means of the Synergy Finder calculation tool, the synergistic effects of 17-AAG and tunicamycin can be demonstrated for distinct Ba/F3 ITD variants, being more pronounced in the Ba/F3-G613E variant.

### 3.2. Impact of 2-Deoxy-D-Glucose on Signal Transduction, Apoptosis, and Metabolic Activity of Ba/F3 Cells Expressing FLT3-ITD Variant 598/599 or G613E

Considering the potential toxicity profiles of glycosylation inhibitors, subsequent investigations of 2-deoxy-D-glucose and its impact on Ba/F3 cells that were expressing either the 598/599 or G613 FLT3-ITD variant were performed. Figure 2 summarizes the effects of 2-deoxy-D-glucose on downstream signaling, apoptosis, and proliferation in both FLT3-ITD-expressing Ba/F3 cell lines. In detail, 2-deoxy-D-glucose lead to a strong reduction of FLT3 expression that resulted in minimal expression of the FLT3-ITD isoform that was characterized by low glycosylation. The sequential treatment of both Ba/F3 FLT3-ITD model systems with 2-deoxy-D-glucose and 17-AAG was associated with the abolishment of all activated downstream pathways that were investigated. Furthermore, the combined treatment resulted in a further reduction of detectable FLT3 protein and a decrease of MCL-1 expression.

We next analyzed the impact of sequential treatment with 2-deoxy-D-glucose and 17-AAG on apoptosis and proliferation. Single treatment with 17-AAG did not affect the level of apoptotic cells or Ba/F3 metabolism as demonstrated in Figure 2B. In contrast, there was an increase in apoptosis when Ba/F3 FLT3-ITD cells were incubated with 2-deoxy-D-glucose only. Ba/F3 cells expressing the 598/599 or the G613E FLT3-ITD variant showed additive effects following the sequential treatment with 2-deoxy-D-glucose and 17-AAG. These results correlated with data that was obtained from the MTS assay analyzing metabolic activity depending on incubation of Ba/F3 FLT3-ITD 598/599 or G613E cells with 17-AAG, 2-deoxy-D-glucose or its sequential treatment (Figure 2B).

Figure 2C demonstrates the 17-AAG concentration-dependent metabolic activity as determined by the MTS assay for Ba/F3 FLT3-ITD cells (598/599 versus G613E) when they were pre-incubated with 2-deoxy-D-glucose. We can show a gradual and comparable decrease of IC_50_ values for 17-AAG in both Ba/F3 FLT3-ITD model systems while there is a generally a higher sensitivity towards 17-AAG in the FLT3-ITD cell line harboring the G613E variant. Further analyses with the Synergy Finder calculation tool that suggests additional effects of 2-deoxy-D-glucose and 17-AAG in both Ba/F3-ITD cell lines are shown in the Appendix A.

### 3.3. Impact of Tunicamycin or 2-Deoxy-D-Glucose on Signal Transduction, Apoptosis and Metabolic Activity of Human FLT3-ITD-Expressing MOLM13 Cells

Consistently, both tunicamycin and 2-deoxy-D-glucose significantly inhibited glycosylation of FLT3 in MOLM13 cells, leading to a clear reduction of fully glycosylated FLT3 (Figure 3A,B left). In contrast, pre-incubation of FLT3-ITD-positive MOLM13 cells with 17-AAG resulted in a reduction of total FLT3 independent of its glycosylation status. Such an enhanced protein degradation upon treatment with 17-AAG can also be observed for total AKT and total STAT5, respectively. Importantly, protein expression of total ERK is significantly increased when MOLM13 cells were treated with 17-AAG alone or with 17-AAG following pre-treatment with either tunicamycin or 2-deoxy-D-glucose.

Comparing the glycosylation levels of FLT3 in MOLM13 cells, tunicamycin seemed to generate lower levels than 2-deoxy-D-glucose. This might explain the remarkably higher increase of phospho-STAT5 after pre-treatment with tunicamycin.

Analysis of apoptosis in MOLM13 cells following sequential treatment with tunicamycin or 2-deoxy-D-glucose prior to incubation with 17-AAG demonstrates the synergistic effects for the combination of 2-deoxy-D-glucose and 17-AAG while pre-incubation with tunicamycin resulted in a moderate increase in apoptotic cells (Figure 3A,B upper right). In contrast, the detection of MOLM13 cell viability by means of a PrestoBlue assay revealed a more pronounced effect of the sequential treatment with tunicamycin and 17-AAG (Figure 3A center right).

Figure 3A,B (lower right) demonstrates the MTS assay results for MOLM13 cells, analyzing the dose-dependence of 17-AAG after pre-treatment with tunicamycin or 2-deoxy-D-glucose, respectively. We can show an approximately threefold reduction of IC_50_ values for 17-AAG when MOLM13 cells were pre-incubated with either tunicamycin or 2-deoxy-D-glucose.

### 3.4. Comparison of Tunicamycin and VPA in Apoptosis induction and Proliferation Inhibition in Ba/F3 Cells Expressing the FLT3-ITD Variant G613E

Figure 4A demonstrates the impact of tunicamycin pre-treatment on the induction of apoptosis following incubation with 17-AAG in Ba/F3 cells that are stably expressing the FLT3-ITD variant 598/599 or G613E, respectively. In both of the FLT3-ITD-expressing Ba/F3 cell lines, we demonstrated a significant reduction of IC_50_ values for Ba/F3 cells that were pre-incubated with tunicamycin compared to the control (0.1% (*v*/*v*) DMSO). To exclude relevant “off target” effects of tunicamycin, Ba/F3 cells that were stably expressing the activating KRAS mutation G12D were treated under the same conditions. Importantly, there was no relevant induction of apoptosis upon single or sequential treatment of Ba/F3 KRAS G12D cells with tunicamycin or 17-AAG (Figure 4B).

We next examined the effect of pre-incubation of either tunicamycin or valproic acid (VPA) prior to treatment with 17-AAG in Ba/F3 cells that were expressing the FLT3-ITD G613E variant that was located in the TKD1 domain of FLT3. The rationale to evaluate the impact of VPA pre-treatment consisted of the observation that VPA and all-trans retinoic acid (ATRA) confer sensitivity of FLT3-ITD-expressing leukemia cells towards the mTOR inhibitor rapamycin and the hypothesis of potential non-genomic effects of such histone deacetylase (HDAC) inhibitors as VPA.

Induction of apoptosis as indicated by IC_50_ value for 17-AAG was observed at a significant lower concentration of 17-AAG when FLT3-ITD G613E Ba/F3 cells are pre-incubated with tunicamycin (Figure 4C). In contrast, pre-treatment with VPA had no significant impact on 17-AAG susceptibility in this Ba/F3 cell line. These results correlated with the data that was obtained from the MTS assay that was analyzing the metabolic activity of 17-AAG treatment that was dependent on pre-incubation of Ba/F3 FLT3-ITD G613E cells with DMSO, tunicamycin, or VPA, respectively (Figure 4D, IC50_17-AAG_: DMSO control: 172.8 nM, after tunicamycin pretreatment: 72.4 nM, after VPA pre-treatment 211.0 nM).

### 3.5. Specific Effects of VPA Treatment on Ba/F3 Cells Expressing the FLT3-ITD Variant G613E

Treatment of Ba/F3 FLT3-ITD G613E cells with the HDAC inhibitor VPA led to a gradual increase of FLT3-ITD surface expression in a dose range between 1 and 10 mM VPA. This observation correlated with semi-quantitative analysis of the FLT3-ITD expression ratio (150 kDa versus 130 kDa) indicating a significant increase of surface-bound FLT3-ITD isoform at 3 mM VPA (Figure 5A). The time-dependent increase of FLT3-ITD surface expression is demonstrated in Figure 5B indicating no effect after 6 h while there was a twofold increase of FLT3-ITD expression detected by flow cytometry after 24 h incubation of Ba/F3 FLT3-ITD G613E cells with 3 mM VPA. In contrast, acetylated histone H3 can be shown by immunoblotting after 6 h of 3 mM VPA treatment.

To characterize the impact of VPA not only on FLT3-ITD surface expression and the ratio between the fully glycosylated and the immature form of FLT3-ITD we next investigated potential effects of VPA on downstream signaling pathways of FLT3-ITD by immunoblotting using phospho-specific antibodies (Figure 5A,B). Here, VPA treatment of Ba/F3 FLT3-ITD G613E was associated with both a concentration-dependent and time-dependent increase of phosphorylated ERK and phosphorylated AKT while there was only a slight activation of STAT5.

To determine the potential mechanisms for reinforced FLT3-ITD surface expression (e.g., due to enhanced protein stability) upon VPA treatment, experiments with cycloheximide (CHX), a potent protein synthesis inhibitor, were performed. In detail, Ba/F3 FLT3-ITD G613E cells were pre-treated with or without 3 mM VPA for 24 h before CHX was added for 0, 120, or 240 min to inhibit protein synthesis. As demonstrated in Figure 5C (upper right), this led to an effective inhibition of both the baseline and VPA induced FLT3-ITD expression. Moreover, after VPA incubation the fully glycosylated FLT3-ITD isoform (150 kDa) had a much less sensitive reaction on CHX treatment (Figure 5C, upper and lower left). Further, quantitative assessment of FLT3-ITD mRNA transcript levels showed no difference between VPA-treated and DMSO-treated Ba/F3 cells (Figure 5C, lower right). 

Global proteome analysis is a powerful tool to explore the deregulation of signaling pathways and quantify changes in the abundance of signaling proteins [39,40]. To explore the potential mechanisms that might explain our observations, we performed whole proteome analyses of Ba/F3 FLT3-ITD cells after treatment with either tunicamycin or VPA. A selection of potentially involved proteins demonstrating relevant changes in their expression levels after incubation with either tunicamycin or VPA is summarized in Table 1.

Importantly, the up-regulation of crucial chaperone proteins, calreticulin and calnexin, that are directly involved in protein glycosylation and protein folding in the ER was demonstrated. This correlated with an increased expression of the HSP90 subunit beta1 and the down-regulation of proteins that mediate transport of proteins to the Golgi complex. Furthermore, decreased expression of important signaling proteins (e.g., MAPK1/2) after exposure to tunicamycin could be observed.

In contrast, a different spectrum of proteins that potentially explained our experimental observations revealed relevant expression changes when Ba/F3 FLT3-ITD cells were treated with VPA. We demonstrated the up-regulation of important signal proteins (e.g., PKC delta) or a reduced expression of NEDD4 that was potentially involved in the degradation of the membrane receptors.

### 3.6. Impact of VPA and Rapamycin on FLT3-ITD Signaling

We next explored the cellular effects of VPA and rapamycin on the FLT3-ITD-dependent down-stream signaling in Ba/F3 G613E and MOLM13 cells (Figure 6A). Single treatment with VPA was associated with an increased phosphorylation of AKT. Phosphorylation of ERK was much more pronounced following incubation of Ba/F3 G613E cells with VPA, while ERK expression was significantly reduced upon VPA treatment. In contrast, the phosphorylation status of STAT5 was not affected in both of the FLT3-ITD-expressing cell lines despite STAT5 expression being markedly down-regulated. Single treatment with rapamycin resulted in almost complete inhibition of phosphorylated AKT in both of the FLT3-ITD cell lines and can also be observed when Ba/F3 G613E or MOLM13 cells were co-incubated with VPA and rapamycin. Surprisingly, rapamycin was associated with an increase of phosphorylated STAT5 that persisted after the combined treatment despite significantly lower expression of STAT5.

The cell viability analysis of MOLM13 cells (Figure 6B) demonstrated a significant effect of both incubation with VPA alone or combined treatment with VPA and rapamycin (DMSO 100% ± 6%, VPA alone 59% ± 7%, rapamycin 34% ± 6%, VPA plus rapamycin 21% ± 6%, *p* < 0.0001).

To explore the impact of ATRA on the effect of VPA and rapamycin we performed cell viability analyses of Ba/F3 598/599 and Ba/F3 G613E cells, respectively (Figure 6C). While single treatment with either VPA or rapamycin had no significant effect on cell viability of both of the FLT3-ITD-expressing cell lines, combined treatment with VPA and rapamycin resulted in a significant decrease of cell viability in Ba/F3 598/599 cells (DMSO 100% ± 5%, VPA alone 101% ± 16%, rapamycin 101% ± 12%, VPA plus rapamycin 56% ± 9%, *p* < 0.0001) and Ba/F3 G613E cells (DMSO 100% ± 5%, VPA alone 85% ± 10%, rapamycin 85% ± 17%, VPA plus rapamycin 56% ± 16%, *p* < 0.001). We further demonstrated that the additional treatment with ATRA had no additional effect as compared to the combined treatment with VPA and rapamycin (Ba/F3 598/599 VPA/rapamycin/ATRA 48% ± 8% and Ba/F3 G613E VPA/rapamycin/ATRA 44% ± 10%, respectively.

## 4. Discussion

The treatment of FLT3-ITD-positive AML with FLT3-TKIs regularly results in “on target” resistance that is mediated by the acquisition of additional point mutations in the FLT3 gene. Thus, there is an unmet medical need to develop new pharmacological approaches to overcome such resistance mechanisms. One promising strategy is reflected by the modulation of intracellular localization of FLT3-ITD that enables combination treatments that address the distinct downstream signaling pathways of the constitutively activated receptor tyrosine kinase.

The inhibition of N-glycosylation by either tunicamycin or 2-deoxy-D-glucose mediates a strong increase of phosphorylated STAT5 as described previously. The significant reduction of phosphorylated STAT5 upon sequential treatment with tunicamycin and 17-AAG can be at least in part explained by a pronounced degradation of the FLT3-ITD compartment that is localized in the GA or ER that can preferentially activate STAT5 [21,23,28].

Considering that STAT5 can directly up-regulate MCL-1, representing an important anti-apoptotic protein in AML cells, 17-AAG-mediated degradation of FLT3-ITD might be an effective strategy to overcome resistance in FLT3-ITD-harboring AML [26,41]. Furthermore, phosphorylation of 130 kDa FLT3-ITD by PIM-1 enhances FLT3-ITD-driven STAT5 signaling by promoting the up-regulation of PIM kinases that are able to interact with the mTORC1/MCL-1 pathway [42,43].

The significant decrease of phosphorylated AKT at lower 17-AAG concentrations compared to the pre-treatment with tunicamycin might be a result of the reduction of fully glycosylated FLT3-ITD that is localized within the cell membrane and is supposed to be most relevant for the downstream activation of AKT. A similar mechanism might underly the observation that the phosphorylation of ERK is more susceptible to inhibition by 17-AAG after pre-treatment with tunicamycin.

Our analyses of Ba/F3 cell lines that were expressing two distinct FLT3-ITD variants located within the JMD or TKD1 domain revealed a similar effect of sequential treatment with 2-deoxy-D-glucose and 17-AAG as compared to incubation with tunicamycin followed by 17-AAG. Importantly, the sensitivity towards both single treatment with 2-deoxy-D-glucose or sequential incubation with different concentrations of 17-AAG appears to depend on the subtype of the FLT3 variant that is expressed in Ba/F3 cells. This let us hypothesize that the location or individual sequence of the FLT3-ITD motif may contribute to the susceptibility to these compounds [10].

Pre-treatment with one of the N-glycosylation inhibitors is clearly associated with an increase of phospho-ERK. In contrast to recent publications, this observation suggests a switch of intracellular signaling that is characterized by a reduction of phospho-AKT and phospho-ERK in favor of STAT5 activation only when FLT3-ITD-harboring cells are treated with tunicamycin or 2-deoxy-D-glucose.

This aberrant activation of ERK following incubation with tunicamycin or 2-deoxy-D-glucose is of high relevance for future treatment strategies when considering glycosylation inhibitors that target FLT3-ITD. Such a “paradox activation” of ERK has been described for other targeted treatment approaches and might be circumvented by the sequential application of MEK inhibitors [44].

Analysis of signaling pathways and the induction of apoptosis after sequential treatment with tunicamycin and 17-AAG reveals a potential impact of such a treatment approach. Inhibition of N-glycosylation leading to a reduced translocation of FLT3-ITD to the cell membrane confers a higher sensitivity to HSP90 inhibitors, representing a proposed common mechanism for certain oncoproteins. Importantly, HSP family members are characterized by high protein expression in FLT3-ITD-positive AML cells [28,45].

To our best knowledge, we demonstrated, for the first time, that the HDAC inhibitor VPA is able to increase the 150 kDa isoform of FLT3-ITD in a time-dependent and concentration-dependent manner. This is accompanied by characteristic changes of 150 kDa versus 130 kDa FLT3-ITD as detected by immunoblotting and can be shown by a significant increase of FLT3-ITD surface expression when analyzed by flow cytometry. This is associated with enhanced phosphorylation of AKT and ERK upon VPA treatment of FLT3-ITD-expressing AML cells as it might be expected following translocation of FLT3-ITD to the cell membrane.

The observation that VPA treatment has no effect on FLT3 mRNA expression supports the hypothesis of a non-genomic mechanism of VPA-induced membrane stabilization of FLT3-ITD in Ba/F3-ITD cells. Proteome analysis reveals the up-regulation of PKC delta and a reduced expression of the ubiquitin protein ligase E3 (NEDD4). The enhanced expression of PKC delta may explain the increase of AKT phosphorylation while the down-regulation of NEDD4 can confer a higher half-time of cell surface proteins including tyrosine kinase receptors [46,47].

By means of proteome analysis of Ba/F3 FLT3-ITD cells that were treated with tunicamycin, we can demonstrate the up-regulation of chaperone proteins that are critically involved in FLT3-ITD stabilization in the ER compartment. This can result in an even more pronounced retention of FLT3-ITD thus increasing the susceptibility of FLT3-ITD-expressing AML cells towards the inhibition of HSP90 [48]. Importantly, FLT3-ITD itself can enhance the expression of GRB94 and HSP90 beta1, propagating a positive feedback loop in the ER compartment [49]. Furthermore, our observation of down-regulated proteins that are necessary for RAS or MAPK signaling let us hypothesize that such effects of tunicamycin on differential protein expression contributes to the altered downstream signaling when FLT3-ITD-positive cells are treated with N-glycosylation inhibitors.

The observation of increased FLT3-ITD surface expression upon VPA treatment let us investigate FLT3-ITD-mediated down-stream signaling in Ba/F3 and MOLM13 cells. The enhanced activation of the anti-apoptotic AKT/mTOR pathway following incubation with VPA is supposed to contribute to the high susceptibility of FLT3-ITD-expressing cells towards the mTOR inhibitor rapamycin. The differential activation of ERK signaling after VPA treatment, as demonstrated for the Ba/F3 G613E and MOLM13 cells, respectively, let us hypothesize that phospho-proteome analysis of AML blasts might represent a promising strategy to identify those patients who might benefit from other combination therapies (e.g., VPA plus MEK inhibitors). While Cai and colleagues demonstrated an additional effect of VPA and rapamycin exclusively by co-incubation with ATRA, we elucidated the impact of ATRA treatment as well. Interestingly, there was no additional effect on cell viability by a triple combination of VPA, rapamycin, and ATRA as shown for Ba/F3 cells that were expressing either the 598/599 or the G613E FLT3-ITD variant.

## 5. Conclusions

Taken together, we can demonstrate that pharmacological strategies allocating FLT3-ITD to different cellular compartments might represent a promising approach for sequential targeting of distinct FLT3-ITD downstream signaling pathways. In consideration of the worst clinical outcome of patients that are harboring the FLT3-ITD location within the TKD domain, patient-specific ITD motifs might influence the effect of the pharmacological strategies that are described in this manuscript.

## Figures and Tables

**Figure 1 cells-10-02992-f001:**
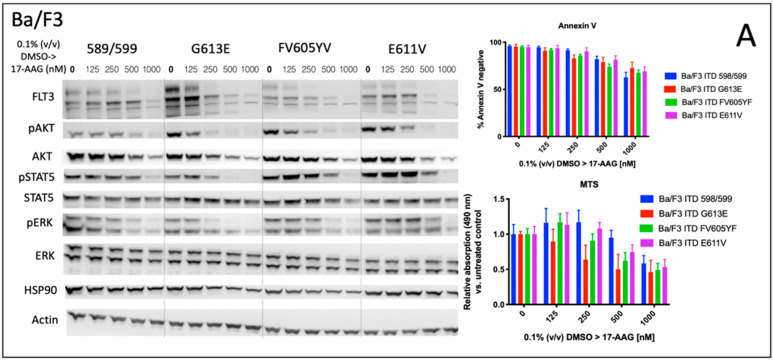
Signal transduction, metabolic activity, and apoptosis after treatment of Ba/F3 cell lines expressing four distinct FLT3-ITD variants with 17-AAG after pre-incubation with tunicamycin. Immunoblotting analysis of signaling pathways (left) after sequential incubation of four Ba/F3 cell lines stably expressing different FLT3-ITD variants with 1 μg/mL tunicamycin and 17-AAG. (**A**) Ba/F3 cells were incubated with 0.1% (*v*/*v*) DMSO for 24 h followed by treatment with 17-AAG at the indicated concentrations for 24 h. (**B**) the incubation was started with 1 μg/mL tunicamycin for 24 h prior to treatment with 17-AAG at indicated concentrations for a further 24 h. Whole cell protein lysates were subjected to SDS-PAGE and immunoblotting to assess the expression and phosphorylation status of the signaling proteins. For loading control, the levels of beta-Actin were detected. Representative blots of the three independent experiments are shown. Corresponding assays indicating metabolic activity (MTS) and apoptosis (Annexin V) are illustrated in the right section of all Ba/F3 FLT3-ITD cell lines. The data for MTS assays are shown as relative absorption compared to untreated controls. Apoptosis is demonstrated by Annexin V-negative cells indicating viable, non-apoptotic Ba/F3 cells. Graphs are shown as the mean ± SD that were obtained from three independent experiments. Statistical significance between each Ba/F3 cell lines comparing DMSO vs. tunicamycin group was calculated using paired *t*-tests. *p* values of Student’s *t*-test, raw data, as well as IC_50_ values are shown in Appendix A.

**Figure 2 cells-10-02992-f002:**
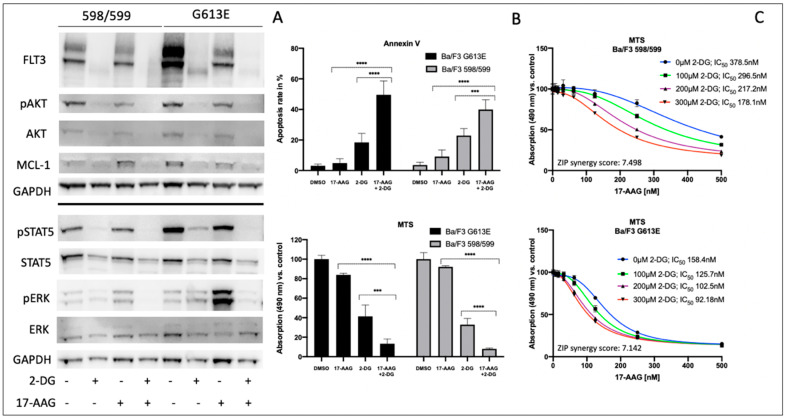
Impact of 2-deoxy-D-glucose on signal transduction, metabolic activity, and apoptosis of FLT3-ITD-expressing Ba/F3 cells (variant 598/599 and G613E). Ba/F3 cells stably expressing distinct FLT3-ITD were treated with 400 μM 2-deoxy-D-glucose or 150 nM 17-AAG or sequentially as indicated. The incubation time for single compounds was 32 h for 2-deoxy-D-glucose and 24 h for 17-AAG. In the combination experiments, the cells were pre-treated with 2-deoxy-D-glucose for 8 h prior to incubation with 17-AAG for further 24 h. (**A**) cell lysates were subjected to SDS-PAGE to analyze the expression and phosphorylation status of signaling proteins. For loading control, levels of GAPDH were detected. Representative blots of three independent experiments are shown. Section (**B**) provides the corresponding data for detection of apoptosis by Annexin V staining (upper part) and metabolic activity by the MTS assay (lower part), respectively. Apoptosis is indicated by Annexin V-positive cells. The data of the MTS assay are demonstrated as relative absorption compared to untreated control. (**C**) 17-AAG concentration-dependent metabolic activity of both Ba/F3-ITD variants after 8 h pre-incubation with indicated concentrations are demonstrated. 17-AAG was added for further 24 h. IC_50_ values are indicated for 17-AAG depending on different concentrations of pre-treatment with 2-deoxy-D-glucose. The calculation of potential synergy of distinct drug combinations as well as biological replicates are described in the Section 3 and illustrated in the Appendix A. A Student’s *t*-test was used for calculating the statistical significance (**** *p* < 0.0001, *** *p* < 0.001, ns not significant).

**Figure 3 cells-10-02992-f003:**
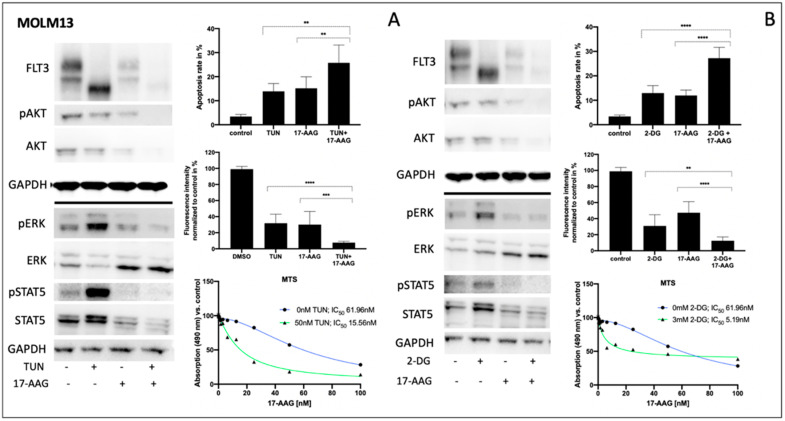
Impact of tunicamycin or 2-deoxy-D-glucose on signal transduction, apoptosis, and proliferation of human FLT3-ITD-expressing MOLM13 cells. Human FLT3-ITD expressing MOLM13 cells were pre-treated for indicated combinations either with 50 nM tunicamycin (**A**) or 3 mM 2-deoxy-D-glucose (**B**) for 24 h after 75 nM 17-AAG was added to the wells for further 24 h. The protein lysates were subjected to immunoblotting to assess changes of indicated signalling proteins. For loading control, levels of GAPDH were detected. The corresponding apoptosis rates are shown after the cells were stained with Annexin V and subsequently analyzed by flow cytometry (upper right of **A** and **B**). For same conditions, a PrestoBlue viability assay was carried out and the dataset that is normalized to the control is illustrated (center right **A** and **B**). IC_50_ values are indicated for 17-AAG dependence on different concentrations of pre-treatment with tunicamycin and 2-deoxy-D-glucose (lower right, **A** and **B**). Student’s *t*-test was used for calculating the statistical significance (**** *p* < 0.0001, *** *p* < 0.001, ** *p* < 0.01, ns not significant). The data for the MTS assays are shown as relative absorption compared to untreated controls. The histograms provide representative data of triplicates that were reproduced in three independent experiments. (mean values and ± SD). The replicates are shown in the Appendix A.

**Figure 4 cells-10-02992-f004:**
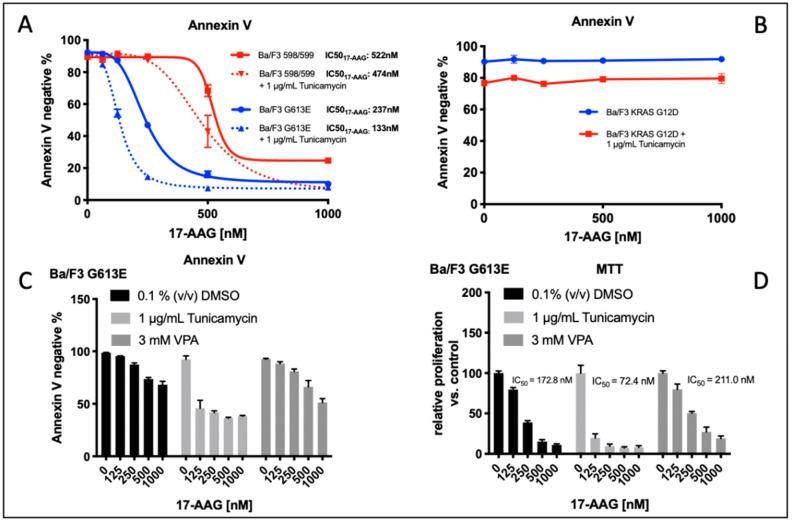
Induction of apoptosis and impact of tunicamycin on the metabolic activity of Ba/F3 cells expressing FLT3-ITD (variant 598/599 and G613E) or an activating mutation of RAS (G12D). Annexin V staining was performed after sequential incubation of different Ba/F3 cell lines. (**A**,**B**), Ba/F3 cells stably expressing FLT3-ITD or KRAS variants were incubated with 0.1% (*v*/*v*) DMSO or 1 μg/mL tunicamycin for 24 h followed by treatment with 17-AAG at the indicated concentrations for 24 h. Apoptosis is demonstrated by Annexin V-negative cells indicating viable, non-apoptotic Ba/F3 cells. IC_50_ values for 17-AAG are shown as mean ± SD that were obtained from three independent experiments. (**B**) Identical conditions were applied for FLT3-ITD-negative Ba/F3 cells that were harboring active KRAS mutation G12D resulting in IL-3-independent growth of Ba/F3 cells. The graphs are shown as mean + SD that were obtained from two independent experiments. (**C**,**D**), Annexin V staining and MTS assay were performed after pre-incubation of Ba/F3-ITD G613E cells with 0.1% (*v*/*v*) DMSO, 1 μg/mL tunicamycin, or 3 mM VPA for 24 h each prior to treatment with the indicated concentrations of 17-AAG for 24 h. Apoptosis is demonstrated by Annexin V-negative cells indicating viable, non-apoptotic Ba/F3 cells. Data for MTS assays are shown as relative absorption compared to the untreated controls. The histograms provide representative data of three independent experiments (mean values and ± SD).

**Figure 5 cells-10-02992-f005:**
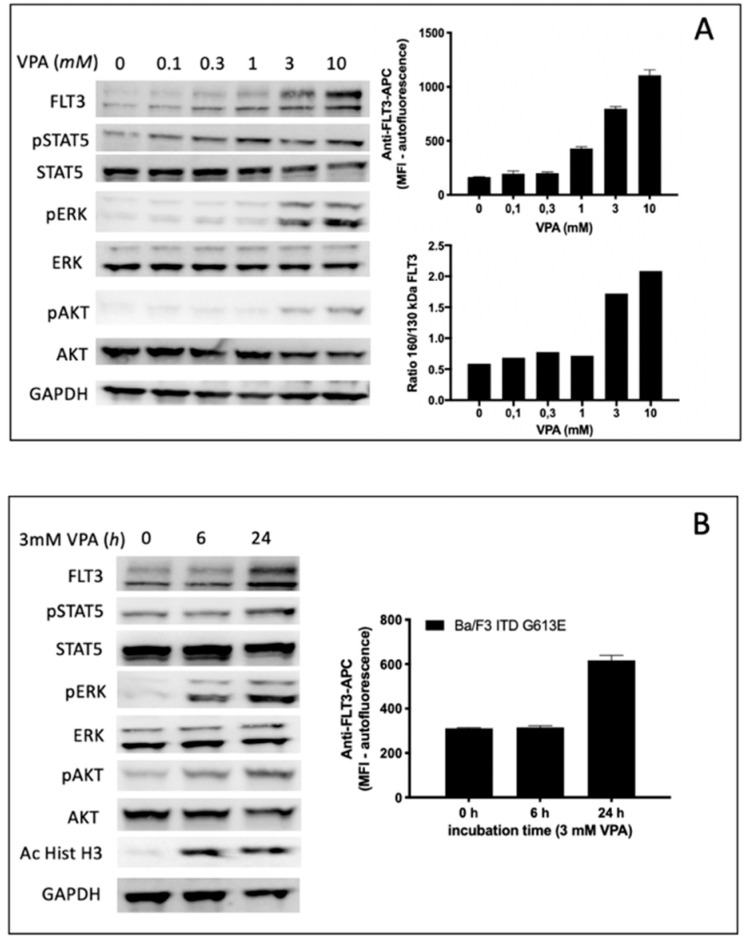
Impact of VPA on FLT3-ITD expression and downstream signaling in Ba/F3-ITD G613E cells. (**A**,**B**) concentration-dependent (**A**) and time-dependent (**B**) expression of surface FLT3-ITD and semi-quantitative analysis of 150 kDa versus 130 kDa isoform of FLT3-ITD. (**A**) Cells were treated for 24 h at the indicated VPA concentrations with subsequent analysis of FLT3-ITD expression by immunoblotting (**A**, left) and flow cytometry (**A**, upper right), respectively. (**B**), cells were treated with 3 mM VPA for 6 h and 24 h, respectively, prior to analysis of FLT3-ITD surface expression by flow cytometry (**B**, right). Immunoblotting was performed for analysis of the downstream signaling pathways. Acetylhistone H3 expression served as a positive control for the VPA effect. For the loading control, the levels of GAPDH were detected. (**A**, left and **B**, left). Densitometric analyses were performed by ImageJ Software. C, Ba/F3 FLT3-ITD G613E cells were pre-incubated with or without 3 mM VPA for 24 h, afterwards 30 μg/mL cycloheximide (CHX) was added for the indicated time intervals. The protein lysates were subjected to immunoblotting (**C**, upper left) and the ratio of 150 kDa versus 130 kDa isoform of FLT3-ITD that was dependent on VPA and CHX treatment was calculated by ImageJ Software (**C**, lower left). FLT3-ITD surface expression was measured by flow cytometry (**C**, upper right). The expression of FLT3-ITD mRNA levels after 24 h incubation of Ba/F3 FLT3-ITD G613E cells with 0.1% (*v*/*v*) DMSO or 3 mM VPA FLT3-ITD were quantified by qPCR (**C**, lower right). The graphs are shown as the mean ± SD from three independent experiments.

**Figure 6 cells-10-02992-f006:**
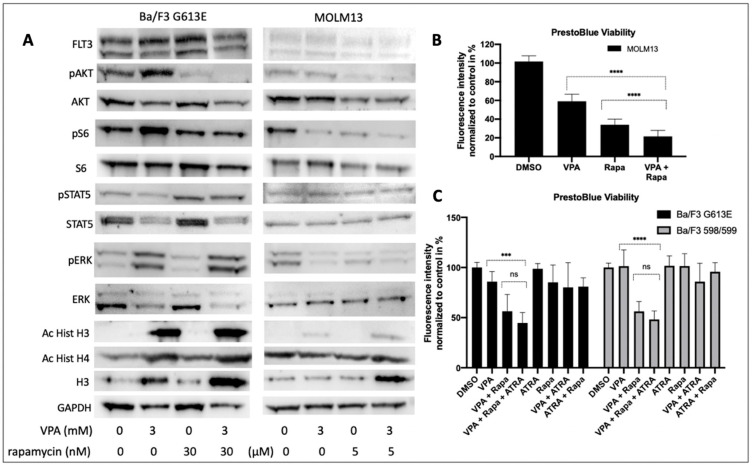
Induction of VPA and rapamycin on FLT3-ITD the downstream signaling and cell viability in Ba/F3 G613E and MOLM13 cells. Ba/F3 cells that were stably expressing FLT3-ITD G613E variant or MOLM13 cells were treated with 3 mM VPA alone, either 30 nM (Ba/F3) or 5 nM (MOLM13) rapamycin or a combination of both compounds as indicated. (**A**) The cell lysates were subjected to SDS-PAGE to analyze expression and phosphorylation status of signaling proteins. For the loading control, the levels of GAPDH were detected. The representative blots of two independent experiments are shown. (**B**) MOLM13 cells were incubated with VPA and/or rapamycin as indicated prior to viability analysis by PrestoBlue assay. (**C**) Ba/F3 cells that were either expressing the 598/599 or the G613E variant were incubated with VPA and/or rapamycin as indicated with or without 1 mM ATRA. the cell viability analyses were performed by PrestoBlue assay. The graphs are shown as mean ± SD from three independent experiments. Student’s *t*-test was used for calculating the statistical significance (**** *p* < 0.0001, *** *p* < 0.001, ns not significant).

**Table 1 cells-10-02992-t001:** Differential protein expression of Ba/F3-ITD (G613E) cells after incubation with either 1 μg/mL tunicamycin or 1 mM VPA for 48 h. Whole proteome analysis was performed in triplets by liquid chromatography–tandem mass spectrometry (LC-MS). All indicated fold changes were statistically significant (*p* < 0.05) while red and green colored illustrating down- and upregulation, respectively.

	Protein	“Fold Change”	Function of Protein
**Tunicamycin**	Calreticulin	3.11	Chaperone complex in endoplasmatic reticulum
	HSP90beta1	2.53	
	Calnexin	1.63	
	RAS GTPase protein binding protein2	0.55	Signal transduction
	MAPK1/2	0.51	
	Endoplasmic reticulum-Golgi intermediate compartment protein1	0.45	Protein transport ER to GA
**VPA**	Protein kinase C delta type	1.69	Signal transduction
	Nuclear factor NF-kappa B p105 subunit	1.64	Transcription factor
	Eukaryotic translation initiation factor 4E binding protein 1 (4E-BP1)	0.59	Translation
	Cyclin-dependent kinase4 (CDK4)	0.66	Cell cycle
	E3 ubiquitin-protein ligase NEDD4	0.64	Degradation of membrane receptors

## Data Availability

Not applicable.

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
