# Peer review of "Modulation of FLT3-ITD Localization and Targeting of Distinct Downstream Signaling Pathways as Potential Strategies to Overcome FLT3-Inhibitor Resistance"

_cells, 2021, doi:10.3390/cells10112992_

Round 1

Reviewer 1 Report

This manuscript use AML cell lines to model FLT3-ITD variants to demonstrate the effect of alternative drugs on downstream signaling pathways, and potentially over come FLT3-inhibitor resistance. All data are generated in cell lines, which reduced the significance. The authors basically use Western blot to detect down stream signaling pathway proteins upon drug treatment. Experimental design is simple. Results conclusion are not well explained.  Proteomics data can be analyzed more thoroughly. Suggest major revision.

Author Response

Dear Reviewer,   on behalf of the authors I’m very grateful for your efforts and time spending for detailed reviewing our manuscript. Please let me give some explanations regarding your important points. 
  1. Cell models: Our main intention was to investigate the shifting of FLT3-ITD with consecutive changes of downstream signaling in a kind of mechanistically way. Therefore, the Ba/F3 and MOLM13 cells seemed to be the most accurate models, especially regarding potentially dependence on FLT3-ITD localization. Experiments with primary cells or in mouse models weren’t planned for this project and would - at least in our eyes - go beyond the scope. 
  2. Conclusion/Results section: Of course we will revise this part comprehensively in order to clarify our key messages even better. 
  3. Proteomics: This analysis primarily served as an addition approach to our comprehensive western blot analysis  with the primary intention of hypothesis generation or supporting potential mechanisms of action (e.g. up-regulation of HSP90 via tunicamycin). Analysis was performed in three biological replicates. We could add the p-values to the table, change its visualization or take this part to the supplementary material. 
In consideration of the necessity to perform additional experiments we would be most grateful to be allowed of submitting the revised version until 29-Oct-2021.
  Best regards Maximilian Fleischmann On behalf of the authors. 

Reviewer 2 Report

This manuscript investigates the effects of N-glycosylation inhibitors (tunicamycin or 2-deoxy-D-glucose) or the histone deacetylase inhibitor valproic acid (VPA) on FLT3-ITD localization and downstream activity. The ultimate goal is to provide an effective combination therapy addressing either N-glycosylation and the heat shock protein 90 (HSP90) by 17-AAG or targeting the PI3K/AKT/mTOR pathway by rapamycin after treatment with VPA on different AML FLT3-ITD variants. Following are some suggestions that would promote the quality of the manuscript.

1: An combination index of 17-AAG and 2-DG treatments is suggested to be revealed in the figure 2, figure 3 and figure 6.

2: VPA is a pan HDAC inhibitors and therefore several epigenetic modifications are being activated. Histone H3 and H4 are typically acetylated after VPA exposure. An additional H4 histone acetylation is suggested to be involved in figure 6.  

Author Response

Dear Reviewer,

on behalf of the authors I’m very grateful for your efforts and time spending for detailed reviewing our manuscript. Please let me give some explanations regarding your important points. 

  1. We would like to mention that synergy calculations of 17-AAG and 2-DG for figure 2 and 3 were performed and are illustrated in the supplement. If you suggest to add them in the figures of the main manuscript we can do so. As well we can deliver a synergy score for VPA and rapamycine for figure 6.  
  2. We just ordered an acetyl. H4 western blot antibody and will perform this assay as soon as possible.  

In consideration of the necessity to perform additional experiments we would be most grateful to be allowed the submission of the revised version until 29-Oct-2021.

Best regards

Maximilian Fleischmann

On behalf of the authors. 

Round 2

Reviewer 1 Report

The authors answered my questions.